# Relationship between Risk Factors Related to Eating Disorders and Subjective Health and Oral Health

**DOI:** 10.3390/children9060786

**Published:** 2022-05-26

**Authors:** Eun-Ha Jung, Mi-Kyoung Jun

**Affiliations:** 1Department of Dental Hygiene, College of Medical Convergence, Catholic Kwandong University, Gangneung 25601, Korea; jeunha725@cku.ac.kr; 2Sae·e Dental Clinic, 109-8, Songwon-ro, Jangan-gu, Suwon 16294, Korea

**Keywords:** adolescent, eating disorder, eating behavior, oral health, body mass index

## Abstract

This study examined the factors related to eating disorders (ED) and the relationship between ED and subjective health or subjective oral health in adolescents. The 46,146 adolescents (age 12–18 years) who participated in the Korea Youth Risk Behavior Web-based Survey were selected, including those who had attempted to lose weight within the past 30 days during the survey period. The variables included were eating behavior, BMI, body image subjective health, and subjective oral health. The weight-loss method was divided into two groups (regular exercise, RE, and eating disorder, ED). The data were analyzed using the Rao-Scott χ^2^ test and logistic regression analysis. The adolescents with an obese body image had a lower risk of ED (OR = 0.75, 95% CI 0.38–1.49) than adolescents with a very thin body image. Adolescents with ED had a higher risk of a poor subjective health assessment (OR = 2.32, 95% CI 1.85–2.91). On the other hand, they had a lower risk of a poor subjective oral health assessment (OR = 0.89, 95% CI 0.71–1.12). ED is closely associated with eating behavior, BMI, body image, oral health behavior, subjective health, and subjective oral health in Korean adolescents.

## 1. Introduction

Eating disorders (EDs) are psychogenic disorders associated with several serious psychological and physical complications and are among the nine most serious problems affecting adolescents [1]. According to the World Health Organization (WHO) International Classification of Diseases, EDs are classified into three types: Anorexia, Bulimia, and Binge eating disorders, which feature abnormal eating habits and pain or extreme anxiety about weight control, body shape and weight [2,3].

EDs are common in women, usually starting during puberty [4]. Adolescent EDs typically last for more than five years, indicating that EDs persist throughout adolescence and the transitional period to early adulthood. As such, prolonged eating disorders can have a significant impact on the growth and development of adolescence [5,6]. Although the incidence and severity of EDs may vary over time and between individuals, early diagnosis and treatment can significantly increase the chances of disease recovery [7]. On the other hand, few seek treatment because most ED patients tend to deny or hide their disease and avoid professional help [8]. Therefore, identifying the symptoms and factors associated with the onset of EDs in adolescence, an important period of biopsychosocial development in an individual’s life, is important for effective prevention and early intervention strategies.

People with EDs are at increased risk for mental health problems and additional health problems, including diabetes, loss of menses in females, heart failure, metabolic disorders, and cardiovascular and endocrine disorders [9]. In addition to these systemic problems, EDs are closely related to oral diseases, such as dental erosion, dental caries, dry mouth, salivary gland edema, and periodontal disease. [10]. Of these, the most obvious and well-explained oral complication is gastric acid-induced dental erosion during self-induced vomiting [11]. A recent review reported that individuals with EDs were five times more likely to have dental erosion than healthy individuals. Moreover, those with self-induced vomiting had a seven times higher risk [12]. Dry mouth was reported to be related to behaviors caused by anorexia nervosa [13], but there is a difference of opinion in the literature regarding the effects of EDs on the major oral diseases, gingivitis, and periodontal disease [14,15]. Although there are some related prior studies [16,17], studies evaluating the oral symptoms occurring in EDs are insufficient. Because oral symptoms differ according to the specific behaviors associated with various types of eating disorders, it will be necessary to identify the symptoms and factors to reduce oral and medical complications associated with eating disorder behaviors.

Therefore, this study examined the relationship between eating EDs and subjective health and oral health in adolescence to provide primary data for establishing a system to combat EDs. 

## 2. Materials and Methods

### 2.1. Research Design and Research Tool

This cross-sectional study is a secondary analysis of descriptive research using the 15th Korea Youth Risk Behavior Web-based Survey (KYRBS, 2019) to find the factors that can affect EDs and to investigate the relationship between EDs and subjective health or oral health among Korean adolescents. The KYRBS was approved by the Korea Disease Control and Prevention Agency before the investigation (Approval No. 117058). The study was conducted according to the relevant guidelines and regulations. The present study received approval (Approval No. P01-202108-21-004) from the Public Institutional Review Board Designated by the Ministry of Health and Welfare. In this paper, an evaluation was done of the quality of reporting for cross-sectional studies using the STROBE guideline.

### 2.2. Description of Variables

The 15th KYRBS had 15 parts and 105 questions in an anonymous, self-reported format. In this study, the questions related to the subject’s demographic characteristics (five items), eating behavior (six items), obesity and weight control (five items), oral health behaviors and symptoms (10 items), subjective health (one item), and subjective oral health (one item) were extracted. Subsequently, each variable was reclassified according to the purpose of this study.

### 2.3. General Characteristics

Gender, age, subjective body type recognition, body mass index (BMI), academic grades, and economic status and residential type were selected as the analytical variables. Gender was divided into “male” and “female”. The age ranged from 12 to 18 years old. For classifications according to subjective academic performance and domestic economic status over the most recent 12 months, “top”, “mid-upper”, “mid”, “middle-low” and “low” were reclassified into “high”, “middle” and “low”.

### 2.4. Eating Behavior

The variables related to the eating behavior included six items as follows; fruit consumption more than once a day, milk consumption more than once a day, soft drinks more than three times a week, caffeinated beverage consumption more than three times a week, water intake more than 600 mL a day, and educational experience concerning eating habits and nutrition in the past year.

### 2.5. Obesity and Losing Weight Attempts

Before analysis, the BMI was calculated as the weight (kg)/height (cm) using the data collected from adolescents. The calculated BMI was divided into five levels according to the Korean Society for the study of obesity. Subjective body type recognition was divided into very thin, slightly thin, normal, slightly overweight, and obese. In KYRBS, subjects had to select how to control their weight if they had tried to lose weight in the last 30 days. Thus, the weight control attempts and weight loss methods (regular exercise, vomiting after eating) were included as variables. Among the weight loss methods, subjects who tried to lose weight with regular exercise were defined as RE, and those who tried to lose weight by vomiting after eating were defined as eating disorder (ED).

### 2.6. Oral Health Behavior and Oral Disease Symptom

There are 10 items regarding oral health behavior and oral disease symptoms. Oral behavior consisted of the number of times of brushing per day, experience of dental visits and oral health education in the past year, and whether or not brushing occurred before bedtime. Oral diseases consisting of broken teeth, pain during chewing, tingling and throbbing, bleeding gingiva, pain in the tongue or inside the cheek, and symptoms of bad breath were included as variables.

### 2.7. Subjective Health and Subjective Oral Health

Subjective health and subjective oral health variables were selected. The values were divided into five levels originally and these were divided into three levels, “healthy”, “normal”, and “unhealthy” for analysis.

### 2.8. Statistical Analysis

Following the KYRBS data analysis guidelines, complex sample data analysis was conducted using the corrections for strata, cluster, weight, and a finite population. All analyses included the use of weighted variables. A Rao-Scott χ^2^ test was conducted to examine the relationship between variables. To identify the characteristics of factors affecting eating disorders in adolescents and to evaluate the effects of eating disorders on subjective health and oral health, variables with *p*-values < 0.05 in univariate analysis were selected as independent variables by correcting for sex, age, and BMI. As a result, multiple logistic regression was performed. Differences with *p*-values <0.05 were considered significant. All data were analyzed by IBM SPSS Statistics program version 21.0 (SPSS, Chicago, IL, USA).

## 3. Results

### 3.1. General Characteristics

As a result of the general characteristics of the participant, a total of 46,146 adolescents were included, of which 22,672 (49.1%) were male, and 23,474 (50.9%) were female (Table 1). Significant differences in gender, age, subjective academic performance, and home economy were observed in the participants who responded that they had tried to lose weight within the last one month or had not (*p* < 0.001). Among the 19,167 participants who had tried to lose weight, the response rate of female students was approximately 1.5 times higher than that of male students. On the other hand, the current residence type of the target group was mostly living with a family (approximately 94.9%), and there was no difference in characteristics according to the type of residence (*p* = 0.114).

### 3.2. Eating Behavior According to Weight Loss Methods

Table 2 lists the difference between the participants’ eating behavior and whether they had tried to lose weight, such as ‘Regular exercise’ or ‘Vomiting after eating’. In all eating behavior variables, there was a significant difference according to the effort to lose weight, such as ‘Regular exercise’ or ‘Vomiting after eating’ (*p* < 0.05). In the RE or ED group, 21.7% and 21.2%, respectively, responded that they ate fruit more than once a day, while the proportion of participants who responded that they consumed milk more than once a day was 26.4% and 23.0%, respectively. In addition, the proportion of participants who responded that they consumed carbonated beverages four or more times a week was 34.9% and 47.1%, respectively, and for consumed caffeinated beverages four or more times a week 11.9% and 24.4%, respectively. As a result of the analysis of the eating behaviors and nutritional education over the past year, 52.7% of the participants currently were performing ‘Regular exercise’, and 43.9% of the participants who tried to lose weight through ‘Vomiting after eating’ had some educational experience.

### 3.3. Subjective Health and Subjective Oral Health According to Weight Loss Methods

Table 3 lists the difference between the participants’ subjective health and subjective oral health and whether they had tried to lose weight, such as ‘Regular exercise’ or ‘Vomiting after eating’. In subjective health, in adolescents who had tried to lose weight through regular exercise, the proportion of those who perceived that they were healthy was high, whereas approximately 53.4% of adolescents who had tried to lose weight by vomiting after eating recognized that they were currently unhealthy. In the case of subjective oral health, approximately 40.1% and 3.1% of adolescents who had tried to lose weight through regular exercise or ‘Vomiting after eating’ perceived that they were healthy, respectively (*p* < 0.001).

### 3.4. Oral Health Behavior and Oral Disease Symptoms According to Weight Loss Method

Table 4 lists the difference between the participants’ oral health behavior and symptoms and whether they had tried to lose weight, such as ‘Regular exercise’ or ‘Vomiting after eating’. All oral health behavior and symptoms variables showed significant differences in whether weight loss was attempted, such as ‘Regular exercise’ or ‘Vomiting after eating’ (*p* < 0.001). In the RE group, the frequency of toothbrushing at least three times a day (54.9%), visiting the dental clinic for oral examination at least once in the past year (67.7%), and toothbrushing before sleep (87.3%), showed a high rate. In addition, the ED group who had a frequency of brushing at least three times a day (58.7%), visiting the dental clinic for oral examination at least once in the past year (66.5%), and toothbrushing before sleep (79.2%) showed a higher rate than participants who tried other methods. The RE group had experienced more ‘tooth fractures’ during the past year.

### 3.5. Factors Related to Weight Loss Attempts

Table 5 lists the influences of body type recognition, eating behavior, and oral health behavior and symptoms on the groups that attempted to lose weight by regular exercise or vomiting after eating according to the participant’s gender, age, and adjusted BMI. 

As a result of body type recognition, those who perceived themselves as slightly overweight or obese were 1.60 and 1.36 times more likely to try to lose weight through regular exercise, respectively. On the other hand, the group using vomiting after eating had a 0.86 times and 0.75 times lower score, respectively (*p* < 0.001) than those who did not. 

As a result of the eating behavior, adolescents who did not consume carbonated drinks or caffeinated beverages more than three times a week were 0.86 times and 0.98 times less likely to try to lose weight with regular exercise, respectively, but the possibility of trying to lose weight by vomiting after eating was 1.52 times lower and 1.72 times higher (*p* < 0.001). 

An analysis of the relationship between oral health behaviors and symptoms showed that weight loss was attempted with regular exercise 0.82 times and 0.78 times less in those who brushed their teeth three times a day and those who did not brush their teeth before sleep, respectively (*p* < 0.001). Adolescents who had experienced symptoms of tooth fracture during the past year were 1.15 times and 1.35 times more likely to try to lose weight through regular exercise or vomiting after eating, respectively (*p* < 0.01). Adolescents who complained of tingling & throbbing, gingiva pain & bleeding, pain of the tongue & intraoral side were significantly more likely to attempt weight control by vomiting after eating (*p* < 0.01).

### 3.6. Factors Related to Subjective Health and Subjective Oral Health

Table 6 lists the influence of body type recognition, eating behavior, oral health behavior and symptoms, and ED on subjective health and oral health according to the participants’ gender, age, and BMI adjusted.

As a result of body type recognition, the risk of recognizing that subjective health was poor was 1.16 times higher when obesity was perceived than when very thin was perceived. The probability of recognizing that subjective oral health was poor was 1.49 times higher (*p* < 0.001). As a result of eating behavior, in the case of consuming carbonated beverages three or more times a week, the risk ratio that subjective health was poor was 1.06 times higher, and the risk ratio that subjective oral health was poor was 1.23 times higher (*p* < 0.001).

An analysis of the effect of oral health behavior and symptoms on subjective health and subjective oral health showed that the risk of affecting subjective health and subjective oral health was 1.03 times and 1.39 times higher, respectively, when brushing was not performed more than three times a day (*p* < 0.001). In the case of not tooth brushing before sleep, the subjects were 1.13 times and 1.41 times more likely to respond that their subjective health and subjective oral health were not good (*p* < 0.001), respectively. In the case of subjective oral symptoms, an experience of tooth fracture, pain during chewing, tingling and throbbing, gingiva pain and bleeding, or bad breath in the past year, the probability of recognizing that subjective oral health was bad was significantly high. Among these, tingling and throbbing had the strongest influence at 2.18 times (*p* < 0.01).

An analysis of the effect of ED on subjective health showed that the risk of recognizing that subjective health and subjective oral health were bad was low when weight loss was attempted with regular exercise. By contrast, the risk of weight loss attempt by vomiting after eating was 2.32 times higher (*p* < 0.001) (*p* < 0.001). In subjective oral health, when participants tried to lose weight with regular exercise and vomiting after eating, the risk of recognizing that subjective oral health was poor was low (*p* < 0.05).

## 4. Discussion

Adolescence is a term that refers to the period between childhood and youth, and adolescents undergo various physical and mental changes [18]. It is a period in which the importance of appearance is recognized as a social factor in developing the values, attitudes, and skills necessary for social participation [19]. Recently, a social environment that emphasizes appearance has developed in Korea. As a result, the interest in and pressure on appearance in adolescence is increasing. Accordingly, many adolescents try to lose weight to manage their appearance [20]. Although there are many ways to lose weight, ED in adolescents is becoming a serious problem. Therefore, this study analyzed various factors that could affect ED as a problem in Korean adolescents.

As a result, 46,146 (80.5%) of the adolescents who participated in the study had attempted to lose weight, of which 39.4% were male students and 60.6% female students (Table 1, *p* < 0.001). The results confirmed that female students were more interested in losing weight. This suggests that women have higher expectations of a thin appearance, which is ideally required in modern society [19,21]. In other words, the social and psychological pressure towards skinny bodies may have led to weight loss attempts. As a basis for this, it can be compared with the BMI results investigated in this study. The BMI was calculated based on the weight and height information, and each value was divided into five stages. As a result, 10,279 subjects (approximately 53.6%) had attempted to lose weight, even though they were underweight or of normal weight. (Table 1, *p* < 0.001). The cause of this is presumed to be the influence of cognitive distortion regarding their weight, which is a problem in adolescents [22,23]. As a result of checking whether or not to lose weight according to subjective body type recognition, approximately 60.8% of adolescents who answered ‘I appear to have gained weight’ were not gaining weight compared to 42.8% of students who were ‘overweight’ or ‘obese’ based on the BMI. However, many students subjectively perceived that they had gained weight and attempted to lose weight. Hence, cognitive distortions about weight are becoming a global problem. There are also reports of cases where the incorrect recognition of body shape leads to extreme weight control, leading to death [24,25]. Because the BMI index reflects only height and weight, there are limitations in that it does not perfectly reflect the individual’s actual body shape. On the other hand, efforts are needed to narrow the gap between objective body status and subjective cognition. Therefore, forming a healthy body image to solve eating disorders and other health-related problems in adolescents is required.

One of the greatest problems among adolescents is limiting nutritional intake by vomiting after eating to lose weight [26]. Therefore, this study attempted to identify the dietary habits of adolescents suspected of ED by trying to lose weight through vomiting after eating. As a result, there were significant differences between the subjects’ eating behavior and whether they attempted to lose weight. The eating behavior patterns of adolescents who tried to lose weight through regular exercise and those who lost weight by vomiting after eating were similar (Table 2). On the other hand, in the case of girls, the probability of trying to lose weight by vomiting after eating was 1.62 times higher than for boys. Furthermore, the possibility of trying to lose weight by regular exercise was 0.59 times less than boys. Considering this result, ED problems in girls need to be considered more carefully. The risk of ED was higher in girls, who were 0.59 times less likely than men to try to lose weight through regular exercise (Table 5, *p* < 0.001). ED appear to have become more common among younger females during the period when icons of beauty (e.g., contestants at beauty contests) have become thinner, and women’s magazines publish significantly more articles on weight loss methods [21]. As mentioned above, erroneous values and social perceptions of body image can harm the physical development of adolescents. Nevertheless, it is positive that the eating behavior patterns of the RE and ED groups were similar (Table 2). Therefore, this problem will likely be improved if a plan is prepared to help people realize that healthy weight control is possible through proper nutrition and regular exercise.

The eating behavior of the target group attempting to lose weight was analyzed in detail. To the question ‘Do you drink soft drinks more than three times a week?’, approximately 34.9% of the RE group answered ‘yes’, whereas 47.1% of the ED group answered ‘yes’ (Table 2). Thus, the rate of intake of soft drinks was higher in the ED group. Compared to the results of more than 50% of adolescents consuming soft drinks four or more times a week in a previous study, this is a relatively positive result [27]. Nevertheless, soft drinks were consumed relatively frequently in the ED group. Adolescents who consumed soft drinks three or more times a week were 1.52 times more likely to have ED (Table 5, *p* < 0.001). In the case of soft drinks, the possibility of tooth erosion has already been reported in several studies. In general, tooth erosion due to frequent vomiting in ED patients has also been reported [28,29]. Therefore, it is necessary to consider the effects of soft drinks intake in adolescents on ED and on dental health. On the other hand, as for the intake of caffeinated beverages, 11.9% and 24.4% of both groups consumed these four or more times a week, respectively (Table 2). Although the consumption rate of caffeinated beverages was relatively low, the likelihood of complaining of ED was increased by 1.72 times when caffeine was consumed three or more times a week (Table 5, *p* < 0.001). The side effects caused by caffeine may be considered the cause of these results. Caffeine stimulates the central nervous system and peripheral nervous system, so when consumed in an appropriate amount it has positive effects, such as stimulating nerve activity and reducing fatigue. On the other hand, when taken in high doses, it can cause nervousness, excitement, and insomnia, and can adversely affect patients with digestive, endocrine, and heart diseases or gastric ulcers [30,31]. In other words, although it is limited in identifying the direct cause, it will be necessary to consider the effects of caffeine on eating disorders. In particular, adolescents are quite sensitive to caffeine, which can cause neurological disorders and heart disorders. Therefore, caution is required regarding caffeine intake in adolescents [32,33]. 

In dietary and nutrition educational experience, among the RE group and ED group, 52.7% and 56.1% of the subjects who received education, respectively, showed no significant difference in the ratio between the group (Table 2). On the other hand, in an analysis of whether education affected weight loss methods, the rate of weight loss through regular exercise was 0.81 times lower in subjects who did not receive dietary education (Table 5, *p* < 0.001). By contrast, education did not affect the subjects who tried to lose weight by vomiting after eating (Table 5, *p* = 0.141). Currently, textbooks on eating habits for adolescents officially published by the Korean government include the importance of forming correct eating habits in adolescence, how to eat healthily, and education on food safety and nutrition. Although it contains various contents for consuming healthy nutrition in adolescence, it appears that education did not have a positive effect on correct eating and nutritional intake in adolescents. Adolescents are greatly influenced by the media and peers [34]. Therefore, it will be necessary to develop an appropriate medium to deliver the correct values regarding body shape and to promote changes in social perception.

In relation to subjective health, approximately 74.2% of adolescents who had attempted to lose weight through regular exercise answered that they thought they were healthy, whereas approximately 53.4% of adolescents with ED answered that they were not healthy (Table 3). In the case of the RE group, the probability of responding to ‘subjective health is not good’ was reduced by 0.66 times, whereas the result was increased 2.32 fold in subjects who complained of ED problems. Hence, weight loss method can affect the subjective health of the subject (Table 6, *p* < 0.001). On the other hand, in the RE and ED groups trying to lose weight, 35.4% and 28.9% of the subjects responded that their oral health was healthy, respectively. The RE and ED subjects who answered ‘not good oral health’ were 16.4% and 26.5%, respectively; the proportion of subjects who answered ‘not good oral health’ among adolescents with ED was relatively high (Table 3). The effect of the weight-loss method on the oral health of the subjects was low, and the difference between the RE group and the ED group could not be confirmed (Table 6, *p* < 0.05). Overall, adolescents had less consideration for oral health than for their general health. In particular, oral health is sufficiently closely related to be used as an indicator to detect ED in subjects for the first time. Therefore, it is necessary to confirm the effect of ED on oral health in detail in this study [35,36]. 

The ‘number of brushing per day’, ‘frequency of brushing before bedtime’, and ‘the rate of visiting the dentist at least once in the past year’ were high in the RE and ED groups (Table 4). Nevertheless, their effect on ED could not be confirmed (Table 5, *p* > 0.05). In the case of oral health symptoms, the possibility of ED increased when adolescents had symptoms, such as broken teeth (1.75 times), pain when eating food (1.61 times), throbbing teeth (2.18 times), and gingival bleeding (1.45 times) (Table 6, *p* < 0.01). Many studies have reported that oral problems, such as tooth erosion, decreased salivation, and dental caries, occur in patients with eating disorders. Therefore, it is essential to consider oral health in subjects trying to lose weight by vomiting after eating [12,35]. Nevertheless, the fact that adolescents suffering from EDs do not consider oral health is an area that needs to be improved through accurate identification of the causes and systematic oral health education. In addition, as EDs in adolescence are mentioned as a serious problem, it is urgent to develop and disseminate educational content that comprehensively considers their overall health and oral health. For this, convergence research among various occupational groups, such as dental medical personnel, medical personnel, and nutritionists, should be attempted.

This study is meaningful in that it identifies the factors that can affect EDs, which are emerging as serious problems in adolescents, and the effects of EDs on the subjective health and oral health of adolescents. In addition, it suggests alternatives. Nevertheless, this study was a cross-sectional study using the results of the KYRBS, and there was a limitation in inferring a causal relationship between each variable. These limitations can be overcome by conducting a well-organized follow-up study based on the results confirmed in this study.

## 5. Conclusions

Eating disorders in adolescents are affected by eating habits and oral health behaviors that may influence subjective general health and oral health status. Therefore, various media and systematic institutional measures are needed to improve the factors that can affect eating disorders in adolescents. Considering these study results, increased care about eating disorders could improve adolescents’ general and oral health.

## Figures and Tables

**Table 1 children-09-00786-t001:** General characteristics according to losing weight effort within the last one Month.

Variables	Categories	Tried to Lose Weight		χ^2^	*p*
		No (*n* = 26,979)	Yes (*n* = 19,167)		
Sex	Male	15,120 (56.0)	7552 (39.4)	260.259	0.000
	Female	11,859 (44.0)	11,615 (60.6)		
Age (Years)	12	2109 (7.8)	1613 (8.4)	70.843	0.000
	13	4359 (16.2)	3172 (16.6)		
	14	4520 (16.8)	3355 (17.6)		
	15	4521 (16.8)	3285 (17.2)		
	16	4338 (16.1)	3183 (16.7)		
	17	4689 (17.4)	3056 (16.0)		
	18	2350 (8.7)	1439 (7.5)		
Subjective body type recognition	Very thin	1432 (5.3)	118 (0.6)	825.243	0.000
	A little thin	7569 (28.1)	1113 (5.8)		
	Normal	10305 (37.9)	6279 (32.8)		
	A little overweight	6367 (23.9)	9508 (49.9)		
	Obesity	1306 (4.8)	387 (10.9)		
BMI	Underweight	7425 (28.4)	1459 (8.0)	2436.327	0.000
	Normal	13,153 (51.1)	8820 (49.3)		
	Overweight	2428 (9.6)	3229 (18.1)		
	Obesity	2365 (9.2)	3821 (21.2)		
	High obesity	422 (1.7)	627 (3.5)		
Subjective academic performance	High	10,839 (40.2)	6610 (34.5)	37.073	0.000
	Middle	7979 (29.6)	5973 (31.2)		
	Low	8161 (30.2)	6584 (34.4)		
Household economic status	High	10,358 (38.4)	7416 (38.7)	36.132	0.000
	Middle	13,321 (49.4)	9060 (47.3)		
	Low	330 (12.2)	2691 (14.0)		
Residential type	Family	25,594 (94.9)	18,185 (94.9)	1.951	0.114
	Relative	129 (0.5)	116 (0.6)		
	Live alone	132 (0.5)	136 (0.7)		
	Dormitory	1022 (3.8)	662 (3.5)		
	Child care facilities	102 (0.4)	68 (0.4)		

The values are presented as unweighted number (weighted %). The total numbers of some variables are different due to missing values. Calculated using the general linear model for continuous variables or by Rao-Scott χ^2^ test for each variable.

**Table 2 children-09-00786-t002:** Eating behavior according to the weight-loss methods.

Variables	Categories	Regular Exercise				Vomiting after Eating			
		Yes (*n* = 22,029)	No (*n* = 8295)	χ^2^	*p*	Yes (*n* = 717)	No (*n* = 26,979)	χ^2^	*p*
Fruit consumption (more than one time/day)	No	17,241 (37.0)	6871 (15.1)	86.111	0.000	565 (1.2)	23,547 (51.0)	8.640	0.027
	Yes	4788 (41.0)	1424 (12.5)			152 (1.4)	6060 (52.2)		
Milk consumption (more than one time/day)	No	16,204 (36.5)	6688 (15.3)	183.837	0.000	552 (1.2)	22,340 (50.5)	34.648	0.000
	Yes	5825 (42.5)	1607 (12.1)			165 (1.2)	7267 (53.4)		
Soft drinks consumption (less than three times/week)	No	7690 (35.9)	3093 (14.7)	58.115	0.000	338 (1.6)	10,445 (49.0)	89.733	0.000
	Yes	14,339 (39.0)	5202 (14.5)			379 (1.0)	19,162 (52.5)		
Caffeine drinks consumption (less than three times/week)	No	2619 (38.0)	1058 (15.7)	10.312	0.022	175 (2.5)	3502 (51.2)	108.637	0.000
	Yes	19,410 (37.8)	7237 (14.4)			542 (1.0)	26,105 (51.2)		
Water intake (more than 600 mL/day)	No	3757 (27.2)	2422 (18.1)	870.599	0.000	169 (1.2)	6010 (44.1)	366.441	0.000
	Yes	18,272 (41.2)	5872 (13.5)			548 (1.2)	23,596 (53.4)		
Nutrition and eating behaviors education (during one year)	No	10,424 (35.1)	4478 (15.4)	80.685	0000	402 (1.3)	14,500 (49.3)	95.537	0.000
	Yes	11,605 (40.8)	3817 (13.6)			315 (1.1)	15,107 (53.3)		

The values are presented as unweighted numbers (weighted %). The total numbers of some variables are different due to missing values. Calculated using the general linear model for continuous variables or by Rao-Scott χ^2^ test for each variable.

**Table 3 children-09-00786-t003:** Subjective health and subjective oral health according to the weight loss method.

Variables	Categories	Regular Exercise				Vomiting after Eating			
		Yes (*n* = 22,029)	No (*n* = 8295)	χ^2^	*p*	Yes (*n* = 717)	No (*n* = 26,979)	χ^2^	*p*
Subjective health	healthy	16,348 (40.1)	5120 (12.8)	511.136	0.000	134 (3.1)	21,085 (51.9)	186.365	0.000
	Normal	4409 (33.6)	2318 (18.2)			200 (1.5)	6527 (50.4)		
	Unhealthy	1272 (29.5)	857 (20.4)			383 (0.9)	1995 (46.8)		
Subjective oral health	healthy	7795 (41.7)	2308 (12.8)	220.861	0.000	207 (1.1)	9896 (53.4)	78.660	0.000
	Normal	10,617 (36.8)	4219 (14.9)			320 (1.1)	14,516 (50.6)		
	Unhealthy	3617 (34.1)	1768 (16.7)			190 (1.1)	5195 (49.1)		

The values are presented as unweighted numbers (weighted %). The total numbers of some variables are different due to missing values. Calculated using the general linear model for continuous variables or by Rao-Scott χ^2^ test for each variable.

**Table 4 children-09-00786-t004:** Oral health behavior and oral disease symptoms according to weight loss method.

	Variables	Categories	Regular Exercise				Vomiting after Eating			
			Yes (*n* = 22,029)	No (*n* = 8295)	χ^2^	*p*	Yes (*n* = 717)	No (*n* = 26,979)	χ^2^	*p*
Oral health behavior	Frequency tooth-brushing(more than three time/day)	No	9333 (35.2)	4149 (14.9)	169.960	0.000	312 (1.1)	13,770 (49.0)	119.553	0.000
		Yes	12,096 (40.4)	4146 (14.2)			405 (1.3)	15,837 (53.3)		
	Oral examination in dental clinic (more than one time/year)	No	7105 (36.3)	2841 (14.8)	31.084	0.000	240 (1.2)	9706 (49.8)	21.193	0.000
		Yes	14,924 (38.6)	14.5 (0.2)			477 (1.2)	19,901 (51.9)		
	Tooth-brushing before sleep	No	2634 (32.4)	1303 (16.8)	122.522	0.000	112 (1.4)	3825 (48.1)	169.763	0.000
		Yes	19,229 (38.7)	6899 (14.2)			568 (1.1)	25,560 (51.7)		
	Experience of oral health education (during one year)	No	14,775 (36.0)	6113 (15.2)	228.220	0.000	516 (1.2)	20,372 (50.0)	140.982	0.000
		Yes	7254 (42.7)	2182 (12.8)			201 (1.2)	9235 (54.4)		
		Tooth fracture (during one year)	No	19,763 (37.5)	7478 (14.5)	42.539	0.000	590 (1.1)	26,651 (50.9)	95.183	0.000
	Yes		2266 (41.5)	817 (15.0)			127 (2.4)	2956 (0.8)		
	Pain during chewing (during one year)	No	14,463 (38.6)	5085 (13.9)	46.129	0.000	382 (1.0)	19,166 (51.5)	37.731	0.000
		Yes	7566 (36.5)	3210 (15.8)			335 (1.6)	10,441 (50.7)		
	Tingling & throbbing (during one year)	No	17,394 (38.3)	6185 (13.9)	67.035	0.000	446 (1.0)	23,133 (51.2)	104.377	0.000
		Yes	4635 (36.3)	2110 (16.8)			171 (2.1)	6474 (5.,0)		
	Gingiva pain & bleeding (during one year)	No	18,021 (37.8)	6598 (14.2)	33.655	0.000	480 (1.0)	24,139 (51.0)	112.390	0.000
		Yes	4008 (37.9)	1687 (16.3)			237 (2.2)	5469 (52.0)		
	Pain of tongue & intraoral side (during one year)	No	19,734 (38.2)	7199 (14.2)	47.980	0.000	537 (1.0)	26,396 (51.4)	134.950	0.000
		Yes	2295 (35.2)	1096 (17.1)			180 (2.7)	3211 (49.6)		
	Bad breath (during one year)	No	17,635 (38.5)	6407 (14.3)	41.826	0.000	505 (1.1)	23,537 (51.6)	38.910	0.000
		Yes	4394 (35.5)	1888 (15.8)			212 (1.2)	6070 (49.6)		

The values are presented as unweighted numbers (weighted %). The total numbers of some variables are different due to missing values. Calculated using the general linear model for continuous variables or by Rao-Scott χ^2^ test for each variable.

**Table 5 children-09-00786-t005:** Factors related to the association between body type recognition, eating behavior, oral health behavior and symptoms, and weight loss attempts.

Variables	Categories	Regular Exercise			Vomiting after Eating		
		OR	95% CI	*p*	OR	95% CI	*p*
Sex	Male	Ref			Ref		
	Female	0.59	(0.55–0.63)	0.000	1.62	(1.32–1.99)	0.000
Age		0.94	(0.93–0.96)	0.000	1.75	(1.42–2.17)	0.001
BMI	Normal	Ref			Ref		
	Underweight	0.75	(0.69–0.81)	0.000	0.66	(0.47–0.92)	0.000
	Overweight	1.16	(1.06–1.26)	0.000	0.81	(0.62–1.05)	0.000
	Obesity	1.21	(1.09–1.33)	0.000	0.97	(0.74–1.28)	0.000
	High obesity	1.29	(1.06–1.57)	0.000	1.04	(0.57–1.89)	0.000
Subjective body type recognition	Very thin	Ref					
	A little thin	1.43	(1.24–1.65)	0.000	0.82	(0.48–1.37)	0.000
	Normal	1.75	(1.51–2.03)	0.000	0.71	(0.40–1.24)	0.000
	A little overweight	1.60	(1.36–1.88)	0.000	0.86	(0.48–1.55)	0.000
	Obesity	1.36	(1.11–1.66)	0.000	0.75	(0.38–1.49)	0.000
Fruit consumption (more than one time/day)	Yes	Ref			Ref		
	No	0.81	(0.75~0.87)	0.000	0.93	(0.76~1.14)	0.485
Milk consumption (more than one time/day)	Yes	Ref			Ref		
	No	0.80	(0.75~0.86)	0.000	1.00	(0.82~1.21)	0.968
Soft drinks consumption (less than three times/week)	Yes	Ref			Ref		
	No	0.86	(0.81–0.91)	0.000	1.52	(1.27–1.81)	0.000
Caffeine drinks consumption (less than three times/week)	Yes	Ref			Ref		
	No	0.98	(0.89–1.07)	0.603	1.72	(1.42–2.09)	0.000
Water intake (more than 600 mL/day)	Yes	Ref			Ref		
	No	0.89	(0.84–0.95)	0.000	0.93	(0.75–1.15)	0.507
Experience of nutrition and eating behaviors education	Yes	Ref			Ref		
	No	0.89	(0.84–0.95)	0.000	1.14	(0.96–1.36)	0.141
Frequency tooth-brushing (more than three times/day)	Yes	Ref			Ref		
	No	0.82	(0.78–0.87)	0.000	0.89	(0.74–1.06)	0.187
oral examination in a dental clinic (more than one time/year)	Yes	Ref			Ref		
	No	0.95	(0.89–1.00)	0.068	1.18	(0.98–1.42)	0.090
Tooth-brushing before sleep	Yes	Ref			Ref		
	No	0.78	(0.72–0.84)	0.000	1.08	(0.86–1.36)	0.514
Tooth fracture (during one year)	No	Ref			Ref		
	Yes	1.15	(1.04–1.26)	0.006	1.35	(1.08–1.68)	0.008
Pain during chewing (during one year)	No	Ref			Ref		
	Yes	0.96	(0.90–1.02)	0.156	1.13	(0.94–1.36)	0.206
Tingling & throbbing (during one year)	No	Ref			Ref		
	Yes	0.95	(0.88–1.03)	0.221	1.37	(1.12–1.68)	0.002
Gingiva pain & bleeding (during one year)	No	Ref			Ref		
	Yes	1.04	(0.96–1.12)	0.379	1.32	(1.09–1.60)	0.005
Pain of tongue & intraoral side (during one year)	No	Ref			Ref		
	Yes	0.87	(0.80–0.94)	0.001	1.68	(1.33–2.12)	0.000
Bad breath (during one year)	No	Ref			Ref		
	Yes	0.91	(0.84–0.97)	0.007	1.18	(0.97–1.44)	0.097
Experience of oral health education (during one year)	Yes	Ref			Ref		
	No	0.81	(0.76–0.86)	0.000	1.01	(0.84–1.23)	0.886

OR = Odds ratio; CI = Confidential intervals. Multiple logistic regression analysis.

**Table 6 children-09-00786-t006:** Factors related to the association between body type recognition, eating behavior, oral health behavior and symptoms, eating disorder, and subjective health and oral health.

Variables	Categories	Health			Oral Health		
		OR	95% CI	*p*	OR	95% CI	*p*
Sex	Male	Ref			Ref		
	Female	1.67	(1.53–1.82)	0.000	1.62	(1.32–1.99)	0.403
Age		1.15	(1.13–1.18)		1.07	(1.05–1.08)	0.000
BMI	Normal	Ref			Ref		
	Underweight	0.97	(0.85–1.09)	0.000	1.07	0.99–1.15	0.000
	Overweight	0.95	(0.84–1.07)	0.000	0.87	0.80–0.94	0.000
	Obesity	0.99	(0.86–1.12)	0.000	0.71	0.65–0.77	0.000
	High obesity	1.50	(1.21–1.87)	0.000	0.67	0.55–0.81	0.000
Subjective body type recognition	Very thin	Ref					
	A little thin	0.45	0.39–0.53	0.000	0.67	0.60–0.75	0.000
	Normal	0.31	0.23–0.37	0.000	0.63	0.56–0.72	0.000
	A little overweight	0.47	0.39–0.57	0.000	0.88	0.77–1.00	0.000
	Obesity	1.16	0.92–1.48	0.000	1.49	1.25–1.76	0.000
Fruit consumption (more than one time/day)	Yes	Ref			Ref		
	No	1.22	1.11–1.34	0.000	1.14	1.08–1.21	0.000
Milk consumption (more than one time/day)	Yes	Ref			Ref		
	No	1.17	1.06–1.29	0.001	1.05	0.99–1.11	0.090
Soft drinks consumption (less than three times/week)	Yes	Ref			Ref		
	No	1.06	0.99–1.14	0.077	1.23	1.17–1.30	0.000
Caffeine drinks consumption (less than three times/week)	Yes	Ref			Ref		
	No	1.36	1.24–1.48	0.000	1.10	1.03–1.19	0.006
Water intake (more than 600 mL/day)	Yes	Ref			Ref		
	No	1.20	1.11–1.29	0.000	1.16	1.09–1.22	0.000
Experience of nutrition and eating behaviors education	Yes	Ref			Ref		
	No	1.18	1.09–0.27	0.000	1.03	0.98–1.09	0.203
Frequency tooth-brushing(more than three times/day)	Yes	Ref			Ref		
	No	1.03	0.95–1.11	0.524	1.39	1.32–1.46	0.000
oral examination in a dental clinic (more than one time/year)	Yes	Ref			Ref		
	No	1.13	1.05–1.22	0.001	1.02	0.97–1.08	0.373
Tooth-brushing before sleep	Yes	Ref			Ref		
	No	1.31	1.19–1.43	0.000	1.41	1.32–1.51	0.000
Tooth fracture (during 1 year)	No	Ref			Ref		
	Yes	0.95	0.84–1.08	0.440	1.75	1.62–1.89	0.000
Pain during chewing (during one year)	No	Ref			Ref		
	Yes	1.40	1.30–1.51	0.000	1.61	1.53–1.70	0.000
Tingling & throbbing (during one year)	No	Ref			Ref		
	Yes	1.24	1.14–1.35	0.000	2.18	2.06–2.31	0.000
Gingiva pain & bleeding (during one year)	No	Ref			Ref		
	Yes	1.35	1.25–1.47	0.000	1.45	1.36–1.54	0.000
Pain of tongue & intraoral side (during one year)	No	Ref			Ref		
	Yes	1.51	1.38–1.66	0.000	0.91	0.84–0.98	0.013
Bad breath (during one year)	No	Ref			Ref		
	Yes	1.54	1.42–1.67	0.000	2.06	1.95–2.18	0.000
Experience of oral health education (during one year)	Yes	Ref			Ref		
	No	1.09	0.99–1.20	0.068	0.95	0.90–1.01	0.104
Regular exercise	No	Ref			Ref		
	Yes	0.66	0.58–0.74	0.000	0.89	0.83–0.95	0.001
Vomiting after eating	No	Ref			Ref		
	Yes	2.32	1.85–2.91	0.000	0.89	0.71–1.12	0.046

OR = Odds ratio; CI = Confidential intervals. Multiple logistic regression analysis.

## Data Availability

Not applicable.

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
