# Peer review of "Relationship between Risk Factors Related to Eating Disorders and Subjective Health and Oral Health"

_children, 2022, doi:10.3390/children9060786_

Round 1
Reviewer 1 Report
The manuscript is informative and well constructed. However, the long-term consequences of eating disorders for both general and oral health should be mentioned more detailed in the introduction.
Material and methods are briefly and clearly presented.
Results are well prepared, but expressions of percentages are misleading in the tables 2, 3 and 4.
Discussion should cover less results of this study and should present more comparison of other carried out surveys.
Conclusion could be more comprehensive.
Author Response
Thank you for your careful comment. We tried to revise some content for improving this manuscript. Please find attached the revision note.
Thank you so much.
Reviewer 2 Report
Thank you for asking me to review this manuscript. This is an interesting topic area where more research is required. I have some specific comments about the manuscript but I have a few overarching queries which need to be addressed.
The manuscript needs to be clearer regarding the aims and objectives of the study and why certain participants have been included. I am unsure why you have decided to include those losing weight through regular exercise when the focus seem to be on eating disorders and oral health/ health. To be considered eligible for this aspects of the study did participants have to state they were attempting to lose weight (and then you looked at those who stated they were either losing weight by exercise or vomiting (were there any other ways to lose weight recorded?) Again I am not too sure why you have included exercise if you are interested in relationship of risk factors to eating disorders. As a result you have carried out multiple texts/ comparisons and this needs to be addressed within your manuscript.
More specific comments about the manuscript can be found below
Line 13 you state ‘including those who did not attempt to lose weight within the recent 30 days’ this sentence feels out of place as you early say they are part of a youth risk behaviour web based survey. It seems strange to state including those who did not attempt to lose weight. It makes more sense as you read through the abstract and the paper but maybe you should state before the web based survey section that this was about following weight loss. Why have you divided weight loss into these two groups – it seemed from your title that you were looking at eating disorders (and therefore you would look at those with and without an eating disorder?)
Section 2.5 – why are you interested in those losing weight by exercise vs ED – are you not just interested in ED vs no ED or is the survey only for those who are exercising or have defined ED
Section 2.7 were any validated questionnaire used within this section – I see you divided subjective health/oral health into 3 but what were the original options, more information is needed about the questionnaire.
You state the Ed is based on vomiting but in your introduction you stated under reported and response bias for this area – this needs to be addressed within your manuscript.
Page 5
Line 159 you state ‘approximately 35.4% and 28.9% of adolescents 159 who tried to lose weight through regular exercise or 'Vomiting after eating' perceived that they were 160 healthy, respectively (p <0.001).’ I cant see these percentages within your table – where are these data from (same for all the tables throughout I cannot see where your data in the text corresponds to data in your tables)
Can you also explain how your % are calculated in Table 3 - what are they out of (what total) as they don’t equate to 100% - I realise they are weighted but more explanation is needed here.
Overall this manuscript is looking at a variety of different components, regular exercise and eating disorder and then all of the various behaviours and then subjective health and oral health. You have run multiple tests. I think there needs to be clear aims and objectives which are met with the results and concentrated on in the discussion. Your title and introduction don’t fit as well with your results and discussion
Table 5 and line 187-201 – again I cant see how your numbers match up you state 1.6 times more likely to try and lose weight through exercise if overweight but OR is 1.16 – this carries on throughout where I cant see how the data you describe matches to your data in your table.
Issues with multiple comparisons throughout your data – you have performed multiple tests across both RE and ED – have you accounted for this
Page 10 line 229 you state ‘was bad was low when weight loss’ you rephrase slightly or maybe just add a comma in after ‘bad’ to make this sentence easier to read
Results/Discussion
Consider what results you need to answer your research question – there is a lot of information here as per previous comments
Author Response
Thank you for your review. We revised this manuscript to your comment. Please find the attached file.
